# Characterizing engagement dynamics across topics on Facebook

**Gabriele Etta** [1], **Emanuele Sangiorgio**[2], **Niccolò Di Marco** [3], **Michele Avalle**[1], **Antonio Scala**[4], **Matteo Cinelli** [1], **Walter Quattrociocchi**[1] *

1 Center of Data Science and Complexity for Society, Department of Computer Science, Sapienza Università di Roma, Roma, Italy, 2 Department of Social Sciences and Economics, Sapienza Università di Roma, Roma, Italy, 3 Department of Mathematics and Computer Science, University of Florence, Firenze FI, Italy, 4 ISC-CNR UoS Sapienza, Rome, Italy

* quattrociocchi@di.uniroma1.it

**Data Availability Statement:** Data cannot be shared publicly because the study mainly relies on facebook posts obtained from Crowdtangle which, as it states in https://help.crowdtangle.com/en/articles/4558716-understanding-and-citing-

## Abstract

Social media platforms heavily changed how users consume and digest information and, thus, how the popularity of topics evolves. In this paper, we explore the interplay between the virality of controversial topics and how they may trigger heated discussions and eventually increase users' polarization. We perform a quantitative analysis on Facebook by collecting $\sim 57M$ posts from $\sim 2M$ pages and groups between 2018 and 2022, focusing on engaging topics involving scandals, tragedies, and social and political issues. Using logistic functions, we quantitatively assess the evolution of these topics finding similar patterns in their engagement dynamics. Finally, we show that initial burstiness may predict the rise of users' future adverse reactions regardless of the discussed topic.

## Introduction

The advent of social media platforms changed how users consume information online [1–4]. The micro-blogging features on Twitter and Facebook, combined with a direct interaction between news producers and consumers, have remarkably affected how people get informed, shape their own opinions, and debate with other peers online [5–7]. Over the years, following the business model of social media platforms, news outlets and producers attempted to maximize the time spent by users on their contents [8, 9], giving birth to the concept of *attention economy* [10]. The term refers to the users' limited capability and time to process all information they interact with [11–13]. The transition toward a news ecosystem shaped on social media platforms unveiled patterns in information consumption at multiple scales [14, 15], which contributed to the emergence of the polarisation phenomenon and the formation of like-minded groups called echo chambers [16–18]. Within echo chambers, characterized by homophily in the interaction network and bias in information diffusion towards like-minded peers, selective exposure [19] is a significant driver for news consumption [16]. The combination of echo chambers and selective exposure makes users more likely to ignore dissenting information [20], choosing to interact with narratives adhering to their point of view [15, 21].

Several studies explored the existence of these mechanisms in many topics concerning political elections, public health, climate change, and trustworthiness of the news sources [15,

crowdtangle-data, cannot be shared in CSV format. However, any researcher can require access to CrowdTangle upon request. Our Supporting information files contain all the contents to guide the interested reader in replicating our study with information about CrowdTangle access and data collection methodology.

**Funding:** This study was supported by the 100683 EPID Project "Global Health Security Academic Research Coalition" provided by UK/G7 in the form of funds to WQ, GE, MA, MC [SCH-00001-3391], the SERICS under the NRRP MUR program funded by the European Union - NextGenerationEU in the form of funds to WQ [PE00000014], the project CRESP from the Italian Ministry of Health under the program CCM 2022 granted to WQ, and by the PON project "Ricerca e Innovazione," funded by Ministero dell'Istruzione, dell'Università e della Ricerca, granted to MC. The funders had no role in study design, data collection and analysis, decision to publish, or preparation of the manuscript.

**Competing interests:** The authors have declared that no competing interests exist.

21–29]. Findings indicate neither the topic nor the quality of information explains the users' opinion-formation process. Instead, several studies observed how the virality of discussions can increase the likelihood of inducing polarization, hate speech, and toxic behaviors [30–32], highlighting how recommendation algorithms may have a role in shaping the news diet of users.

Therefore, it is necessary to provide a better understanding of how user interest evolves in online debates. To achieve this goal, we provide a quantitative assessment of the dynamics underlying user interest in news articles about different topics. In this paper, we analyze the engagement patterns produced by $\sim 57M$ posts on Facebook related to $\sim 300$ topics, involving a total of $\sim 2M$ posting pages and groups over a period that ranges from 2018 to 2022. We first provide a quantitative assessment of topics' attention through time, extracting insightful parameters from their engagement evolution. Then, we construct a metric called the Love-Hate Score to estimate the level of controversy associated with a topic using the sentiment of users' engagement, as expressed by the normalized difference between their positive and negative reactions. Our results show that topics are generally characterized by an interest that constantly increases since the appearance of the first post. We find that topics' interactions grow with permanent intensity, even for prolonged periods, indicating how interest is a cumulative process that takes time. We statistically validate this result by comparing parameters across topic categories, discovering no differences in the evolution of the engagement. Indeed, regardless of their category, topics keep users engaged steadily over time, and their lifetime progression seems thus unrelated to its thematic field. Finally, we find that topics with sudden virality tend to occur with more controversial and heterogeneous interactions. In turn, topics with a steady evolution exhibit more positive and homogeneous reaction types. This difference in the sentiment of reactions, and the protracted duration of topics' lifetime, are both upshots consistent with the emergence of selective exposure as a driver of news consumption.

## Materials and methods

This section describes the data collection process, the topic extraction process, the models and the metrics employed in assessing collective attention.

### Overview of the data collection process

The data collection process comprises several parts, as described in Fig 1. We start by creating a sample of news articles from the GDELT event database [33]. Then, we process the articles' text to obtain a set of representing terms. Consequently, we apply the Louvain community detection algorithm [34] on the bipartite projection of the co-occurrence term network to identify the topics of interest. The terms representing these topics will serve as input for collecting posts from Facebook.

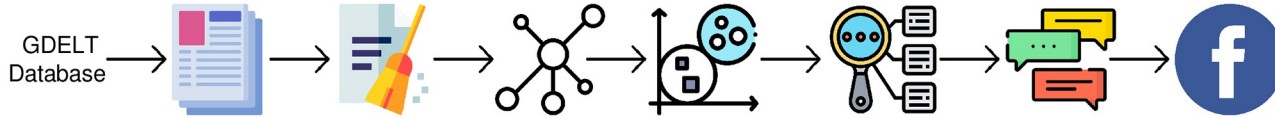

**Fig 1. Summary of the analysis workflow followed in the current study.** News articles are collected from the GDELT Database, and their corpus is extracted, cleaned and analyzed to retrieve the most representing terms. The bipartite projection of the co-occurrence network built upon these terms serves as an input for the Louvain community detection algorithm to identify keyword clusters. Independent labellers then analyze these clusters to identify the subset of words that represent the topic under consideration, which are then used on Crowdtangle to retrieve the Facebook posts relating to those events.

The data collection and analysis process are compliant with the terms and conditions [35] imposed by Crowdtangle [36]. Therefore, the results described in this paper cannot be exploited to infer the identity of the accounts involved.

**News extraction from GDELT.** The GDELT (Global Database of Events, Language, and Tone) Project [37], powered by Google Jigsaw, is a database of global human society which monitors the world's broadcast, print, and web news from nearly every corner of every country in more than 100 languages. It identifies the people, locations, organisations, themes, sources, emotions, counts, quotes, images and events driving our global society every second of every day [38]. We gathered news articles from the GDELT 2.0 Event Database [33], which can store new world's breaking events every 15 minutes and translates the corresponding news articles in 65 languages, representing 98.4% of its daily non-English monitoring volume [33]. The analysis covers a period between 1/1/2018 and 13/5/2022, collecting 50 news articles each week for a total of $\sim 79K$.

**Extracting representative keywords from news articles.** To clean and extract the most representative keywords of each news article, we employed the *newspaper3k* Python package [39]. We initially extracted words from the body of the article, excluding stopwords and numbers. Then, we computed the word frequency $f(w, i)$ for each word $w$ in article $i$. Finally, we sorted words in descending order according to their frequency, keeping the top 10 most frequent words.

**Topic extraction from news article's keywords.** The list of terms with the corresponding news articles can be formalised as a bipartite graph $G = (T, A, E)$ whose partitions $T$ and $A$ represent the set of terms $t \in T$ and the articles $a \in A$ respectively, for which an edge $(t, a) \in E$ exists if a term $t$ is present in an article $a$. By projecting graph G on its terms $T$ we obtain an undirected graph $P$ made up of nodes $t \in T$, which are connected if they share at least one news article.

We perform community detection on the nodes of $P$ by employing the Louvain algorithm [34]. As a result, we obtain a set of clusters $C$, where each cluster $c \in C$ contains a list of keywords that are assumed to be semantically related to a topic. We then asked a pool of three human labellers to select, for each community, from two to three terms they considered the most representative to identify a topic unambiguously.

**Data collection of Facebook posts.** The news articles obtained from the GDELT Event Database do not contain information helpful in estimating the attention they generate online. To include the dimension of user engagement, we employ each topic's set of representative terms to collect Facebook data over a period that goes from 01/01/2018 to 05/05/2022. The data was obtained using CrowdTangle [36], a Facebook-owned tool that tracks interactions on public content from Facebook pages, groups, and verified profiles. CrowdTangle does not include paid ads unless those ads began as organic, non-paid posts that were subsequently "boosted" using Facebook's advertising tools. CrowdTangle also does not store data regarding the activity of private accounts or posts made visible only to specific groups of followers.

The collection process produced a total of $\sim 57M$ posts from $\sim 2M$ unique pages and groups, generating $\sim 8B$ interactions. The result of the data collection process is described in Table 1.

**Table 1. Data Breakdown of the study, including the total amount of news articles and posts collected from GDELT and Facebook respectively, together with the number of topics and the analysis period.**

| News Articles from GDELT | Total Posts from Facebook | Total Interactions | Total Groups and Pages | Number of Topics Collected | Period |
|---|---|---|---|---|---|
| 79 650 | 57 031 026 | 8 015 177 602 | 2 224 430 | 296 | 1/1/2018—5/5/2022 |

**Topic categorization.** To provide a correspondence between topics and their area of interest, we performed a categorization activity under the following labels: Art-Culture-Sport (ACS), Economy, Environment, Health, Human Rights, Labor, Politics, Religion, Social and Tech-Science. Three human labellers carried out the activity to connect topics and categories, choosing as the representative only those categories selected by at least two of the three labellers.

## Metrics

We begin by describing a measure for fitting the cumulative engagement evolution. Then, based on the previous step, we outline an index to evaluate the sharpness of the topic's diffusion. Finally, using Facebook's reactions, we introduce a sentiment score to assess the topic's controversy. A topic-aggregated version of the dataset containing all the metrics defined in this section can be found in the Data Breakdown Section of S1 File.

**Fitting cumulative engagement evolution.** The study of the diffusion of new ideas has been carried on through the years, starting from the Bass diffusion model [40] and then extended to a multitude of topics [41–47], indicating the relevance of s-curves in the analysis of innovation spreading. Therefore, to model the evolution of the engagement received by posts, we fit the cumulative distribution of the overall engagement (i.e., the number of likes, shares and comments) over time employing a function $f_{\alpha,\beta}(t)$, with $\alpha, \beta \in \mathbb{R}$, defined as

$$f_{\alpha,\beta}(t) = \frac{1}{1 + e^{-\alpha(t-\beta)}}. \tag{1}$$

From a mathematical point of view, Eq 1 defines a general sigmoid function that depends on the parameters $\alpha$ and $\beta$. The $\alpha$ parameter represents the slope of the function, describing the steepness of the engagement evolution. On the other hand, $\beta$ is the point at which the function reaches the value 0.5 and quantifies the time required for a topic to reach half its total interactions.

To provide a representation of the impact that $\alpha$ and $\beta$ can have in topic engagement evolution, Fig 2 displays four topics with peculiar configurations. Fig 2a shows a sigmoid in which the high values of $\alpha$ and $\beta$ produce a sharp increment relatively far from $t_0$. Such behaviour corresponds to those topics that require some time before gaining maximum diffusion with the public. Fig 2b instead provides a fit where the sigmoid produces low values for $\alpha$ and $\beta$, resulting in a smoother increment in the proximity of $t_0$ than the one described in Fig 2a. Finally, Fig 2c and 2d provide an example of how two curves that share similar values of $\beta$ parameters can have a different evolution of their increase by slightly modifying the values for $\alpha$ parameter.

**Speed Index.** To provide a measure of how quickly the attention towards a topic reaches its saturation, we define a measure called the Speed Index $SI(f_{\alpha,\beta})$ as

$$SI(f_{\alpha,\beta}) = \frac{\int_0^T f_{\alpha,\beta}(t)dt}{T}. \tag{2}$$

The SI considers the joint contribution of $\alpha$ and $\beta$ parameters, where $T$ represents the time of the last observed value for $f_{\alpha,\beta}(t)$. Note that the $SI$ is the mean integral value of $f_{\alpha,\beta}$, i.e. the normalised area under the curve of $f_{\alpha,\beta}$ (therefore $SI(f_{\alpha,\beta}) \in [0, 1]$). The assumption in the definition of this function relies on the fact that high-speed values are obtained by sigmoids that reach the plateau in a short time, as the behaviour represented in Fig 2b.

**Love-Hate Score.** To quantify the level of controversy that a Facebook post may produce, we define a measure called the Love-Hate (LH) Score. In line with previous works that

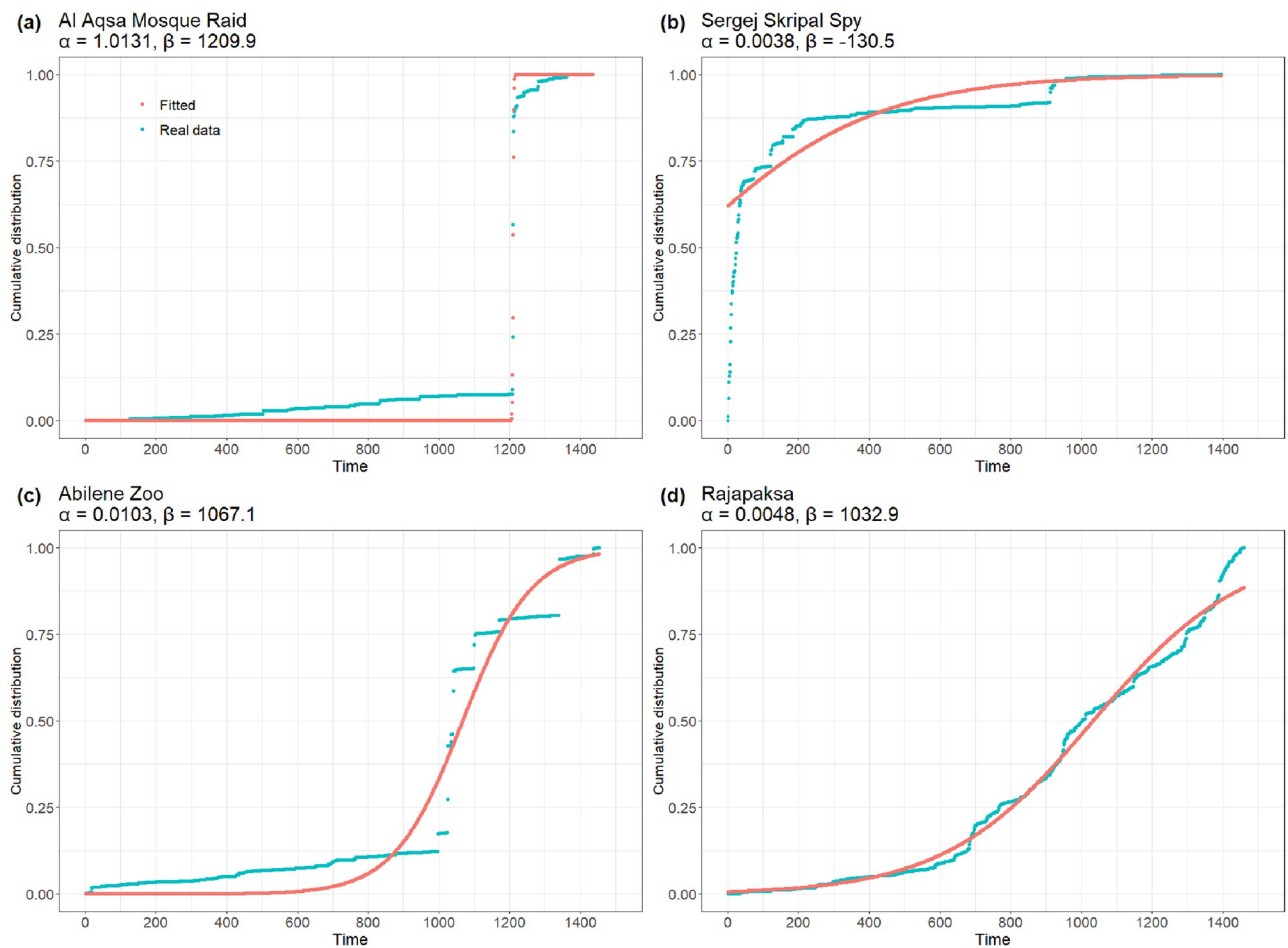

**Fig 2. Representation of a sample of four topics employing their normalized cumulative evolution of engagements and fittings.** The incidence of the $\alpha$ parameter can be observed in the sharpness of the fitting curves. The $\beta$ parameter instead regulates the shift of the function through the $x$ axis: the higher its value, the higher the delay from $t_0$ where the sigmoid produces its increment.

quantified controversy from post reactions [48, 49], we define the LH Score $LH(i) \in [-1, 1]$ as

$$LH(i) = \frac{l_i - h_i}{l_i + h_i},\tag{3}$$

where $h_i$ and $l_i$ are respectively the total number of *Angry* and *Love* reactions collected by a post $i$. A value of *LH* equal to $-1$ indicates that the post received only *Angry* reactions from the users, while a value equal to 1 indicates that the post received only *Love* reactions. Therefore, a value close to 0 reflects the presence of controversy on a post due to a balance of positive and negative reactions.

## Results and discussion

### Quantifying topic engagement evolution

We first provide a quantitative assessment of the the evolution of engagement with topics on social media. To do so, we perform a Non-linear Least Squares (NLS) regression by fitting the sigmoid function $f_{\alpha,\beta}(t)$ to the cumulative engagement gained by each topic.

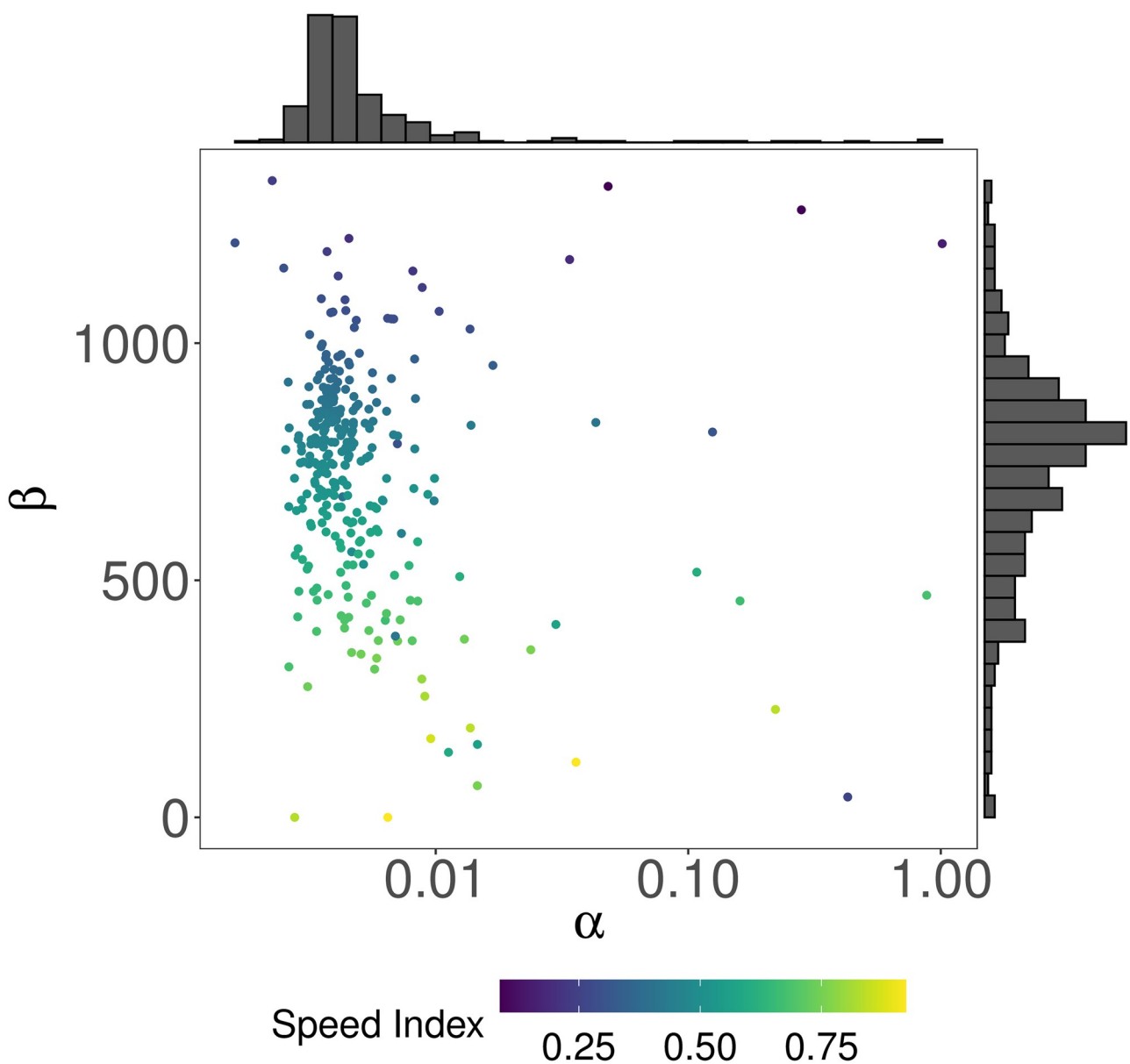

**Fig 3. Joint distribution of $\alpha$ and $\beta$ parameters obtained from the NLS regression for each topic.** We observe that topics are generally characterized by values of $\alpha$ and $\beta$, which explains how user interest in a topic does not increase all of a sudden but is the result of a process that evolves over time.

The distribution of the $\alpha$ parameter provided in Fig 3 describes how the majority of topics have a value of $\alpha$ belonging to the [0, 0.0047] interval. This result demonstrates how user interest in a topic does not suddenly increase but results from a long-term process. Instead, the distribution of the $\beta$ parameter describes a prevalence of topics in the [600, 1000] interval, identifying the tendency of topics to become a matter of interest with some delay w.r.t the first post covering them.

### Evaluating the relationship between topic engagement and controversy

To quantify the interplay between users' interest in a topic and the associated level of controversy, we compute the Spearman correlation between the Speed Index and the LH Score for

each topic. Results from the upper panel of Fig 4 show a general negative tendency of users to react with a negative sentiment when a topic gains engagement faster ($\rho = -0.26$), leaving positive reactions to those topics that require time to obtain maximum diffusion. Results described in the lower panel of Fig 4 provide further characterisation of the interplay between the Speed Index and the LH Score after classifying the topics according to the four most frequent categories analyzed, i.e., Politics, Labor, Human Rights and Health. We observe how the Politics and Health categories have the lowest correlation scores ($\rho = -0.36$ and $\rho = -0.45$), providing an indication of their intrinsic polarizing attitude (see S1 Fig for further details about correlation coefficients). Furthermore, the correlation between $\alpha$ and LH Score produces similar results as with the Speed Index (see S2 Fig for more details).

## Assessing the differences of engagement behaviors across topic categories

To conclude our analysis, we investigate the differences in the evolution of engagement across topic categories. In particular, for each parameter distribution ($\alpha$, $\beta$ and $SI$), we apply a two-tailed Mann–Whitney U test [50] to each pair of parameters. Table 2 provides the percentages of the significant p-values for the four parameters. Due to the necessity to perform multiple tests, we apply a Bonferroni correction to our standard significance level of 0.05, leading to reject the null hypothesis if the p-value $p < 0.001$. Our results show that the resulting p-values from the tests do not lead to rejecting the null hypothesis. Such a result corroborates the hypothesis that, on average, users are characterized by homogeneous engagement patterns that are not influenced by the consumed topic. We further extend the statistical assessment by performing the same test between LH Score distributions of the different categories.

Conversely to engagement evolution results, the topic's category explains differences in the sentiment of reactions in 20% of cases. Such findings reveal that some categories are composed of significantly more negative and controversial topics, indicating how elicited reactions vary according to specific subjects. Understanding that some of them are more prone to induce negative feedback from users could be a proxy to introduce their related topics in the online debate.

## Conclusions

In this work, we perform a quantitative analysis of user interest on a total of $\sim 57M$ Facebook posts referring to $\sim 300$ different topics ranging from 2018 to 2022. We initially quantify the distribution of topics' engagement evolution throughout the analysis. Then, we evaluate the relationship between engagement and controversy. Ultimately, we assess the differences in engagement across different categories of topics. Our findings show that, on average, users' interest in topics does not increase exponentially right after their appearance but, instead, it grows steadily until it reaches a saturation point. From a sentiment perspective, topics that reached a plateau in their engagement evolution right after their initial appearance are more likely to collect negative/controversial reactions, whilst topics which are more steady in their growth tend to attract positive users' interactions. This result provides evidence about how recommendation algorithms should introduce topics adequately since sudden rises in topic diffusion could be related to the reinforcement of polarization mechanisms. Finally, we find no statistical difference between user interest across different categories of topics, providing evidence that, on a relatively large time window, the evolution of engagement with posts is primarily unrelated to their subject. On the contrary, we observe differences in the sentiment generated by topics with different diffusion speed, providing evidence of how people perceive the piece of content they consume online in different ways, according to how suddenly they get exposed to it.

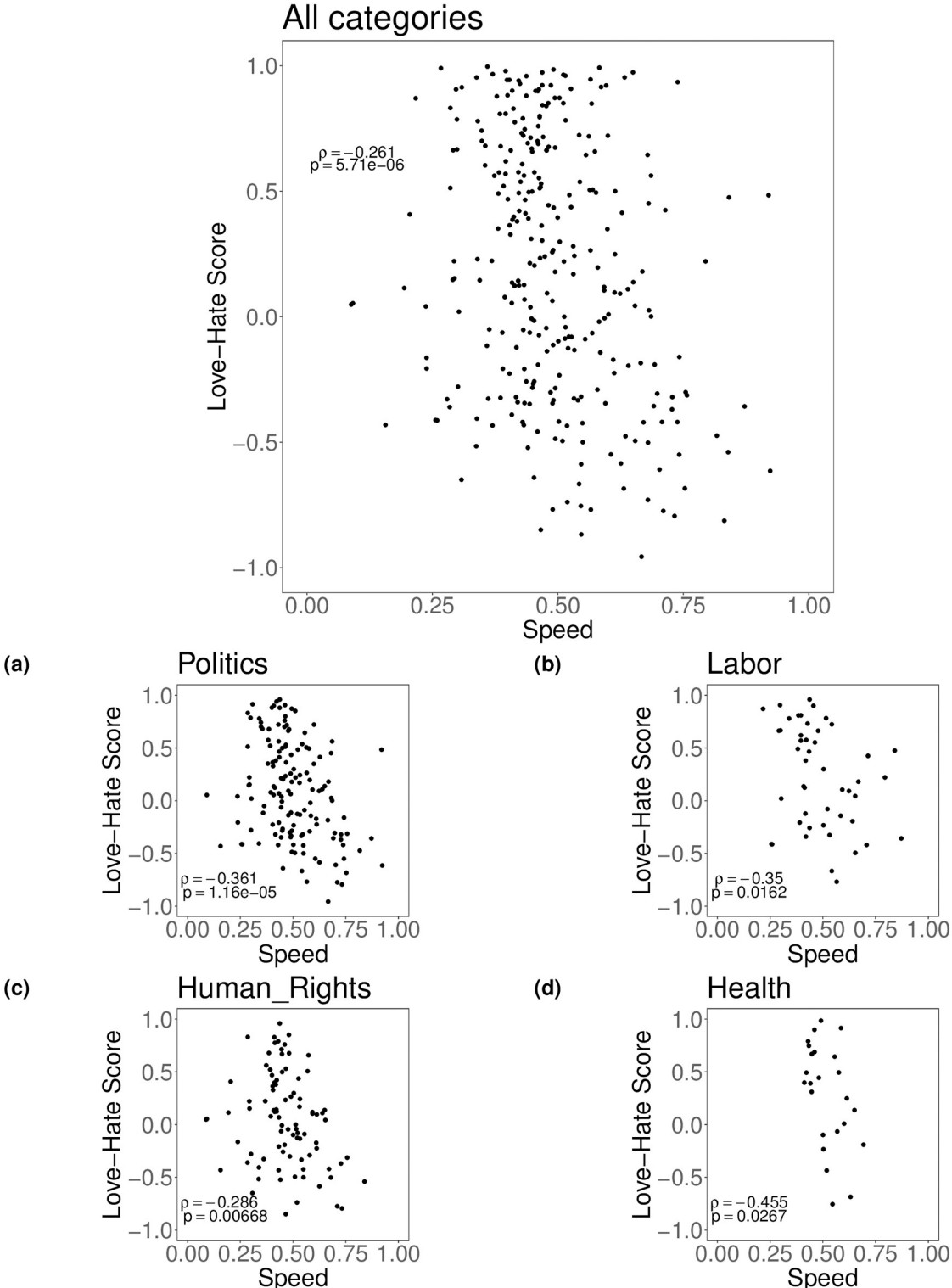

**Fig 4.** Upper panel: correlation between *SI* and *LH* score for each identified topic. Lower panel: correlation between *SI* and *LH* score for the top 4 most frequent topics. Overall, we observe how users react negatively as topics become sharply viral.

**Table 2. Percentage of p-values resulting from the two-sided Mann–Whitney U test between each category employing their $\alpha$, $\beta$, Speed Index and LH Score.**

|  | $\alpha$ | $\beta$ | Speed Index | LH |
|---|---|---|---|---|
| <**0.001** | 2.22% | 0% | 0% | 20% |
| >**0.001** | 97.78% | 100% | 100% | 80% |

Users' interest and engagement evolution in the online debate are both aspects of human behaviour on social media whose underlying dynamics still need to be discovered from an individual point of view. Our findings provide an aggregate perspective of the interplay between major emerging behavioral dynamics and topics' lifetime progression, deepening the relationship between diffusion patterns and users' reactions. Understanding that topics with an early burst in virality are associated with primarily adverse reactions from users may enable the identification of highly polarizing topics since their initial stage of diffusion.

The following study presents some limitations. In data collection, CrowdTangle provides only posts from public Facebook pages with more than 25K Page Likes or Followers, public Facebook groups with at least 95K members, all US-based public groups with at least 2K members, and all verified profiles. These restrictions affected our datasets' sample and our findings' generality. Moreover, we could not access removed posts, groups, and pages, which could have been a meaningful proxy to characterize the attention dynamics of retracted content. Finally, since Crowdtangle does not provide information about users interacting with posts, we cannot assess their engagement from an individual perspective and model the possible relationship between users and topics employing a network approach.

The results obtained in this work may help to better understand how users consume information, improving social media moderation tools by considering both the "life-cycle" of topics and their potential controversy. Indeed, the introduction of the Speed Index and the Love-Hate Score can be exploited to identify in advance topics with the potential to collect considerable interest and generate heated debates quickly. From a news outlet and content creator perspective, understanding that specific topics may reach broader audiences and produce controversial opinions can improve the quality of the communication produced by these two types of authors.

## Supporting information

**S1 Data. This CSV file contains, for each identified topic, the statistics of $\alpha$ and $\beta$ value, the Love Hate Score, the first and last post dates, the topic lifetime (in days), the Speed Index value, the number of posts, total interactions and users posting.**
(CSV)

**S1 File. This file provides the topic aggregated statistics employed in the study.** Moreover, here are provided the figures reporting the correlations between $\alpha$ and LH Score for each topic and the goodness of the fitting procedures.
(PDF)

**S1 Fig. Correlation between $\alpha$ and *LH* score for each identified topic.**
(TIF)

**S2 Fig. Joint distribution of the errors $SE(\hat{\alpha}_i)$ and $SE(\hat{\beta}_i)$ for each topic *i*, whose cumulative curve was estimated by means of $f_{\alpha,\beta}$.** The colour of each point represents the number of posts produced by topic *i*.
(TIF)

## Author Contributions

**Conceptualization:** Gabriele Etta, Emanuele Sangiorgio, Niccolò Di Marco, Michele Avalle, Matteo Cinelli, Walter Quattrociocchi.

**Data curation:** Gabriele Etta, Emanuele Sangiorgio, Niccolò Di Marco, Michele Avalle, Matteo Cinelli, Walter Quattrociocchi.

**Formal analysis:** Gabriele Etta, Emanuele Sangiorgio, Niccolò Di Marco, Matteo Cinelli, Walter Quattrociocchi.

**Funding acquisition:** Walter Quattrociocchi.

**Investigation:** Gabriele Etta, Emanuele Sangiorgio, Niccolò Di Marco, Matteo Cinelli.

**Methodology:** Gabriele Etta, Emanuele Sangiorgio, Niccolò Di Marco, Michele Avalle, Antonio Scala, Matteo Cinelli.

**Project administration:** Gabriele Etta, Emanuele Sangiorgio.

**Resources:** Gabriele Etta, Niccolò Di Marco.

**Software:** Gabriele Etta, Emanuele Sangiorgio.

**Supervision:** Gabriele Etta, Niccolò Di Marco, Walter Quattrociocchi.

**Validation:** Gabriele Etta, Niccolò Di Marco.

**Visualization:** Gabriele Etta, Emanuele Sangiorgio, Niccolò Di Marco.

**Writing – original draft:** Gabriele Etta, Emanuele Sangiorgio, Niccolò Di Marco, Michele Avalle, Antonio Scala, Matteo Cinelli, Walter Quattrociocchi.

**Writing – review & editing:** Gabriele Etta, Emanuele Sangiorgio, Niccolò Di Marco, Michele Avalle, Antonio Scala, Matteo Cinelli, Walter Quattrociocchi.

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
