## [Decision Letter · Decision Letter 0]

13 Mar 2023

PONE-D-22-32836Characterizing Engagement Dynamics across Topics on FacebookPLOS ONE

Dear Dr. ETTA,

Thank you for submitting your manuscript to PLOS ONE. After careful consideration, we feel that it has merit but does not fully meet PLOS ONE’s publication criteria as it currently stands. Therefore, we invite you to submit a revised version of the manuscript that addresses the points raised during the review process. The reviewer raised a few minor issues that should be addressed.

The data availability should be clarified further. In the cited URL of CrowdTangle, it only states that "raw data" cannot be shared publicly. On this basis, it would be possible to share the overall statistics at the topic level as constructed by the author. Please include the data at this aggregated level, or provide a clear reason why this is also not possible,

In addition, there are a few other minor issues that should be improved:

- There are many Figure references missing, this should be corrected.

- There are some causal interpretations for which it is not clear whether such a causal interpretation is actually warranted. For instance, "topics with sudden virality tend to trigger more controversial and heterogeneous interactions", but it is not clear whether the virality actuallly *causes the controversial interactions. There are some other such statements that should be revised.

- It is not really clear whether Speed Index (SI) has much to do with "Speed" or whether this more reflects a quick saturation of attention. Please motivate this measure more clearly.

- For the topic extraction, please make sure that the community detection employed is actually applicable to bipartite networks. There are several possibilities for clustering bipartite networks, and they provide different results from cluster method that are intended to be used on unipartite networks.

We look forward to receiving your revised manuscript.

Kind regards,

Vincent Antonio Traag, Ph.D.

Academic Editor

PLOS ONE

Journal Requirements:

2. In your Methods section, please include additional information about your dataset and ensure that you have included a statement specifying whether the collection and analysis method complied with the terms and conditions for the source of the data.

Reviewers' comments:

Reviewer's Responses to Questions

**Comments to the Author**

1. Is the manuscript technically sound, and do the data support the conclusions?

Reviewer #1: Yes

2. Has the statistical analysis been performed appropriately and rigorously? 

Reviewer #1: Yes

3. Have the authors made all data underlying the findings in their manuscript fully available?

Reviewer #1: Yes

4. Is the manuscript presented in an intelligible fashion and written in standard English?

Reviewer #1: Yes

5. Review Comments to the Author

Reviewer #1: This paper explores the impact of engagement dynamics on Facebook by examining how the increased engagement of controversial topics may trigger heated discussions and increase polarization among users. The authors analyzed 57M posts from Facebook using logistic functions and an s-curve model to assess the evolution of different topics to show that similar patterns in engagement existed for the controversial topics. They measure the sentiment of posts by users' positive and negative reactions and find that topics with sudden virality tend to trigger more controversial and heterogeneous reactions.

The methodology initially uses the GDELT Event Database to create a sample of news articles, and those articles are reduced to top 10 representative words in the article. The authors then apply Louvain community detection on the co-occurrence term networks to identify topics of interest. Those terms were then used as input for the collection of Facebook posts.

The analysis uses sentiment scores to assess the topic’s controversy using Facebook’s reactions. The author's discussion of the difference in the sentiment of reactions between topics with sudden virality and those with a steady evolution is insightful. However, it’s unclear how the authors justify using sentiment scores as a good measure for controversy. In the results section, they define the measure of a Love-Hate score to measure controversy, but it’s slightly confusing from the beginning of the paper where the authors mention using sentiment scores as the measure of controversy. I think the Love-Hate score serves as the ‘sentiment’ score in this case, but it’s not clear. The authors also quantify the relationship between topic resonance and controversy, but again earlier in the paper it’s not clear how the terms of resonance, controversy, and sentiment align with the defined measures in the results section. Possibly add a table clarifying these terms, or just mention in the paper that these terms will be discussed later in the paper in Section X.

The author provides a good overview of the methodology used and the results obtained. However, the paper could benefit from more detailed explanations of some of the terms used, such as "resonance", "controversy”, and sentiment scores in the context these terms are used. It would also be helpful if the author could provide some recommendations for how news outlets or content producers could use this information to improve their content and engagement with users. There have also been other works that should be cited that use a similar approach of s-curves and the sigmoid function in social media such as (Spann, et al., 2022) and Diffusion of Innovations Theory (Bass, 1969).

To be considered for publication, the minor revisions should be applied:

1.) Consider adding a sentence or two explaining the context of topic resonance, controversy, and that sentiment scores are actually defined by the Love-Hate score (if that is indeed the case)

2.) The following references are relevant to the author's work, especially the discussion on diffusion of innovations and s-curves.

- Spann, B., Mead, E., Maleki, M., Agarwal, N., & Williams, T. (2022). Applying diffusion of innovations theory to social networks to understand the stages of adoption in connective action campaigns. Online Social Networks and Media, 28, 100201. https://doi.org/10.1016/j.osnem.2022.100201

- Bass, F. M. (1969). A new product growth for model consumer durables. Management science, 15(5), 215-227.

3.) Consider adding some recommendations for how news outlets or content producers could use this information to improve their content and engagement with users.

Overall, this is a well written and informative manuscript. The author's discussion of the difference in the sentiment of reactions between topics with sudden virality and those with a steady evolution is insightful. I think with the clarifications and additional citations from above, their quantitative analysis will make a valuable contribution to the research community.

6. PLOS authors have the option to publish the peer review history of their article (what does this mean?). If published, this will include your full peer review and any attached files.

Reviewer #1: No

---

## [Author Response · Author response to Decision Letter 0]

4 Apr 2023

Reviewer #1

Consider adding a sentence or two explaining the context of topic resonance, controversy, and that sentiment scores are actually defined by the Love-Hate score (if that is indeed the case)

Authors’ response:

We thank the reviewer for the comment. We addressed this ambiguity by clarifying the usage of resonance, controversy and what Love-Hate score indicates. As result of this process, we introduced the following changes in the text:

In the Introduction section, at Page 2, we changed the following sentence

“We first provide a quantitative assessment of topics' resonance through time, extracting insightful parameters from their engagement evolution. Then, we exploit the obtained parameters by assessing relationships with the sentiment expressed by users through their positive and negative reactions”

To

“We first provide a quantitative assessment of topics' attention through time, extracting insightful parameters from their engagement evolution. Then, we construct a metric called the Love-Hate Score to estimate the level of controversy associated with a topic using the sentiment of users' engagement, as expressed by the normalized difference between their positive and negative reactions. ”

Page 5, in Fitting cumulative engagement evolution section, we changed the sentence “Such behaviour corresponds to those topics that require some time before gaining resonance with the public” into “Such behaviour corresponds to those topics that require some time before gaining maximum diffusion with the public”

Page 6, we changed the content of the Love-Hate Score section, from

“To quantify the level of sentiment that a Facebook post produces, we define a measure of controversy called the Love-Hate (LH) Score $LH (i) \\in \\left[-1,1 \\right]$ as

\\begin{equation}

\\label{eq:lhi}

LH (i) = \\frac{l_i-h_i}{l_i+h_i},

\\end{equation}

where $h_i$ and $l_i$ are respectively the total number of Angry and Love reactions collected by a post $i$. A value of $LH$ equal to $-1$ indicates that the post received only Angry reactions from the users, while a value equal to $1$ indicates that the post received only $Love$ reactions.”

To

“To quantify the level of controversy that a Facebook post may produce, we define a measure called the Love-Hate (LH) Score. In line with previous works that quantified controversy from post reactions [48-49], we define the LH Score $LH (i) \\in \\left[-1,1 \\right]$ as

\\begin{equation}

\\label{eq:lhi}

LH (i) = \\frac{l_i-h_i}{l_i+h_i},

\\end{equation}

where $h_i$ and $l_i$ are respectively the total number of \\textit{Angry} and \\textit{Love} reactions collected by a post $i$. A value of $LH$ equal to $-1$ indicates that the post received only \\textit{Angry} reactions from the users, while a value equal to $1$ indicates that the post received only $Love$ reactions. Therefore, a value close to $0$ reflects the presence of controversy in a post due to a balance of positive and negative reactions.”

We also included two references of papers ([48-49]) that already implemented a similar way of quantifying controversy through post interactions. The papers are:

Beel, J., Xiang, T., Soni, S., & Yang, D. (2022). Linguistic Characterization of Divisive Topics Online: Case Studies on Contentiousness in Abortion, Climate Change, and Gun Control. Proceedings of the International AAAI Conference on Web and Social Media, 16(1), 32-42.

Hessel, Jack; LEE, Lillian. Something's brewing! Early prediction of controversy-causing posts from discussion features. arXiv preprint arXiv:1904.07372, 2019.

We changed the title of the Quantifying topic resonance section, at Page 6, to Quantifying topic engagement evolution. Moreover, in this section we changed its initial part from

“We first provide a quantitative assessment of the topics' resonance on social media. To do so, we perform a Non-linear Least Squares (NLS) regression by fitting the sigmoid function $f_{\\alpha,\\beta} (t)$ to the cumulative evolution of the engagement for each topic.”

To

“We first provide a quantitative assessment of the evolution of engagement with topics on social media. To do so, we perform a Non-linear Least Squares (NLS) regression by fitting the sigmoid function $f_{\\alpha,\\beta} (t)$ to the cumulative engagement gained by each topic.” 

In page 7, we changed the Evaluating the relationship between topic resonance and controversy section name into Evaluating the relationship between topic engagement and controversy. Moreover, in that section we changed the use of adverse with a negative, gain resonance with obtaining maximum diffusion and we changed “To quantify the interplay between users’ interest in a topic and the controversy it produces” to “To quantify the interplay between users’ interest in a topic and the associated level of controversy”

In the Conclusions section, at page 9, we changed the occurrence of “resonance” with “engagement evolution". In the same section, we then changed the second occurrence of “gained resonance” with “reached a plateau in their engagement evolution” and the third one with “diffusion”.

The following references are relevant to the author's work, especially the discussion on diffusion of innovations and s-curves.

- Spann, B., Mead, E., Maleki, M., Agarwal, N., & Williams, T. (2022). Applying diffusion of innovations theory to social networks to understand the stages of adoption in connective action campaigns. Online Social Networks and Media, 28, 100201. https://doi.org/10.1016/j.osnem.2022.100201

- Bass, F. M. (1969). A new product growth for model consumer durables. Management science, 15(5), 215-227.

Author’s response:

We thank the reviewer for pointing out these relevant references that we included in the manuscript. Accordingly to the suggestion, we modified the beginning of Fitting cumulative engagement evolution subsection, at Page 4, from:

“The diffusion of new ideas has been widely studied in the past [40-45], indicating how the logistic function can effectively model the diffusion of innovations.”

To

“The study of the diffusion of new ideas has been carried on through the years, starting from the Bass diffusion model [41] and then extended to a multitude of topics [42-48], indicating the relevance of s-curves in the analysis of innovation spreading.”

Where [41] is the reference to the Bass’ paper and [48] the reference to Spann's work.

 Consider adding some recommendations for how news outlets or content producers could use this information to improve their content and engagement with users.

Author’s response:

We thank the reviewer for this important suggestion. According to this, we added the following part in the Conclusion session, at Page 10:

The results obtained in this work may help to better understand how users consume information, helping social media regulators improve their moderation tools considering both the "life-cycle" of topics and their potential controversy. Indeed, the introduction of the Speed Index and the Love-Hate score can be exploited to identify in advance topics with the potential to collect considerable interest and generate heated debates quickly. From a news outlet and content creator perspective, understanding that specific topics may reach broader audiences and produce controversial opinions can improve the quality of the communication produced by these two types of authors.

Minor Issues addressed by the Editor:

The data availability should be clarified further. In the cited URL of CrowdTangle, it only states that "raw data" cannot be shared publicly. On this basis, it would be possible to share the overall statistics at the topic level as constructed by the author. Please include the data at this aggregated level, or provide a clear reason why this is also not possible.

Author’s response:

We thank the editor for the comment. We provided a CSV file named Topic_Aggregated_Data.csv and a corresponding section in SI, at Page 18, called Data breakdown at topic level. Such a section contains the title of the file (Topic_Aggregated_Data.csv File) and the following caption:

“This CSV file contains, for each identified topic, the statistics of $\\alpha$ and $\\beta$ value, the Love Hate Score, the First and Last Post Dates, the Topic Lifetime (in days), the Speed Index value, the number of Posts, Total Interactions and Users posting.”

In the Metrics section, at Page 4, we added the following sentence to refer to the new dataset:

“A topic-aggregated version of the dataset, containing all the metrics defined in this section, can be found in Section \\ref{S1_File} of SI.”

There are many Figure references missing, this should be corrected.

Author’s response:

We thank the editor for the comment. We reformatted the manuscript by including the figures, thus reinstantiating the original references.

There are some causal interpretations for which it is not clear whether such a causal interpretation is actually warranted. For instance, "topics with sudden virality tend to trigger more controversial and heterogeneous interactions", but it is not clear whether the virality actually *causes the controversial interactions. There are some other such statements that should be revised.

Author’s response:

We thank the reviewer for the comment. We reviewed all the sentences that refer to causal interpretations, modifying them in all those cases for which the result was obtained through correlation analysis instead of casual ones.

According to this purpose, we modified the following sentences:

In the Introduction Section, Page 2, we modified the “Finally, we find that topics with sudden virality tend to trigger more controversial and heterogeneous interaction” to “Finally, we find that topics with sudden virality tend to occur with more controversial and heterogeneous interaction”

In the Evaluating the relationship between topic engagement and controversy, Page 7, we modified the “[...] providing further evidence of their intrinsic polarizing attitude [...]” sentence into “[...] providing an indication of their intrinsic polarizing attitude [...]”

In the Conclusions section, Page 9, we modified the “This result provides evidence about how recommendation algorithms should introduce topics adequately since sudden rises in topic resonance tend to reinforce the polarization mechanisms” into “This result provides evidence about how recommendation algorithms should introduce topics adequately since sudden rises in topic diffusion could be related to the reinforcement of polarization mechanisms”

In the Conclusions section, Page 9, we modified the “On the contrary, we observe differences in the sentiment generated by the different topics, providing evidence of how polarisation drives people to perceive the piece of content they consume online in different ways, according to their framing and system of beliefs.” into “On the contrary, we observe differences in the sentiment generated by topics with different diffusion speed, providing evidence of how people perceive the piece of content they consume online in different ways, according to how suddenly they get exposed to it.”

In the Conclusions section, Page 9, we modified the “Understanding that topics with an early burst in virality are associated with primarily adverse reactions from users sheds light on their tendency to react instinctively to new content. This approach enables the identification of highly polarizing topics since their initial stage of diffusion, by observing the heterogeneity of users' reactions.” into “Understanding that topics with an early burst in virality are associated with primarily adverse reactions from users may enable the identification of highly polarizing topics since their initial stage of diffusion.”

It is not really clear whether Speed Index (SI) has much to do with "Speed" or whether this more reflects a quick saturation of attention. Please motivate this measure more clearly.

Author’s response:

We thank the reviewer for the comment. To remove the ambiguity in the definition of the Speed Index (SI), in the Speed Index Section, Page 5, we replaced the “To model the evolution of a topic by taking into account the joint contribution of α and β parameters” sentence with “To provide a measure of how quickly the attention towards a topic reaches its saturation, we define a measure called the Speed Index”. We therefore changed the explanation of the equation from “where T represents the time of the last observed value for fα,β (t).” to “The SI considers the joint contribution of α and β parameters, where T represents the time of the last observed value for fα,β (t).”

For the topic extraction, please make sure that the community detection employed is actually applicable to bipartite networks. There are several possibilities for clustering bipartite networks, and they provide different results from cluster methods that are intended to be used on unipartite networks.

Author’s response:

We thank the reviewer for the comment. We actually employed community detection on the projection of the bipartite network on the keywords. Therefore, in the Overview of the data collection process section, at Page 2, we changed the “Consequently, we apply the Louvain community detection algorithm [34] on the co-occurrence term network to identify the topics of interest“ sentence to “Consequently, we apply the Louvain community detection algorithm [34] on the bipartite projection of the co-occurrence term network to identify the topics of interest”.

For coherence, we applied the same clarification in the Figure 1 caption, resulting in the following sentence “[...] The bipartite projection of the co-occurrence network built upon these terms serves as an input for the Louvain community detection algorithm to identify keyword clusters. [...]”

Journal Requirements:

Author’s response:

We thank the editor for the comment. We carefully revised the paper in order to meet PLOS ONE’s style requirements, including the filenamings.

In your Methods section, please include additional information about your dataset and ensure that you have included a statement specifying whether the collection and analysis method complied with the terms and conditions for the source of the data

Author’s response:

We thank the editor for the comment. In the Overview of data collection process, at Page 2, we added the following part:

“The data collection and analysis process are compliant with the terms and conditions [35] imposed by Crowdtangle [36]. Therefore, the results described in this paper cannot be exploited to infer the identity of the accounts involved. ”

In your Data Availability statement, you have not specified where the minimal data set underlying the results described in your manuscript can be found. PLOS defines a study's minimal data set as the underlying data used to reach the conclusions drawn in the manuscript and any additional data required to replicate the reported study findings in their entirety.

Author’s response:

We thank the editor for the comment. Due to the previous response, we cannot share a minimal dataset to reproduce the results. Therefore, we changed the Data Availability statement accordingly.

Please review your reference list to ensure that it is complete and correct.

Author’s response:

We thank the editor for the comment. We carefully revised the entire reference list, performing the following changes:

In the Collective memory in the digital age paper, we added the arXiv reference arXiv preprint arXiv:2207.01042

In the Falkenberg22 paper, we added its own title, i.e., Growing polarization around climate change on social media

In the mann1947test paper, we corrected its title into the correct one, i.e., On a Test of Whether One of Two Random Variables Is Stochastically Larger than the Other.

We removed a duplicate about the GDELT Project.

We changed the Acknowledgments section by including all the funding authorities that supported this study. The section now contains the following sentences: “The work is supported by IRIS Infodemic Coalition (UK government, grant no. SCH-00001-3391), 

SERICS (PE00000014) under the NRRP MUR program funded by the European Union - NextGenerationEU, project CRESP from the Italian Ministry of Health under the program CCM 2022, and PON project “Ricerca e Innovazione” 2014-2020.”

---

## [Editor Report · Decision Letter 1]

10 May 2023

Characterizing Engagement Dynamics across Topics on Facebook

PONE-D-22-32836R1

Dear Dr. ETTA,

We’re pleased to inform you that your manuscript has been judged scientifically suitable for publication and will be formally accepted for publication once it meets all outstanding technical requirements.

Kind regards,

Vincent Antonio Traag, Ph.D.

Academic Editor

PLOS ONE

---

## [Editor Report · Acceptance letter]

1 Jun 2023

PONE-D-22-32836R1 

Characterizing Engagement Dynamics across Topics on Facebook 

Dear Dr. Quattrociocchi:

I'm pleased to inform you that your manuscript has been deemed suitable for publication in PLOS ONE. Congratulations! Your manuscript is now with our production department. 

Kind regards, 

on behalf of

Dr. Vincent Antonio Traag 

Academic Editor

PLOS ONE